# Rebound of shelf water salinity in the Ross Sea

Pasquale Castagno [1]*, Vincenzo Capozzi[1], Giacomo R. DiTullio [2], Pierpaolo Falco[1], Giannetta Fusco[1], Stephen R. Rintoul [3,4,5], Giancarlo Spezie[1] & Giorgio Budillon[1]

Antarctic Bottom Water (AABW) supplies the lower limb of the global overturning circulation and ventilates the abyssal ocean. In recent decades, AABW has warmed, freshened and reduced in volume. Ross Sea Bottom Water (RSBW), the second largest source of AABW, has experienced the largest freshening. Here we use 23 years of summer measurements to document temporal variability in the salinity of the Ross Sea High Salinity Shelf Water (HSSW), a precursor to RSBW. HSSW salinity decreased between 1995 and 2014, consistent with freshening observed between 1958 and 2008. However, HSSW salinity rebounded sharply after 2014, with values in 2018 similar to those observed in the mid-late 1990s. Near-synchronous interannual fluctuations in salinity observed at five locations on the continental shelf suggest that upstream preconditioning and large-scale forcing influence HSSW salinity. The rate, magnitude and duration of the recent salinity increase are unusual in the context of the (sparse) observational record.

---

[1] Dipartimento di Scienze e Tecnologie, Università degli Studi di Napoli "Parthenope", Centro Direzionale, Isola C4, 80143 Napoli, Italy. [2] Grice Marine Laboratory, University of Charleston, 205 Fort Johnson Road, Charleston, SC 29412, USA. [3] CSIRO Oceans and Atmosphere, Hobart, Tasmania, Australia. [4] Centre for Southern Hemisphere Oceans Research, Hobart, Tasmania, Australia. [5] Australian Antarctic Program Partnership, Hobart, Tasmania, Australia. *email: pasquale.castagno@uniparthenope.it

Antarctic Bottom Water (AABW) is the most voluminous water mass in the deep ocean[1] and is the primary source of oxygen to the abyss[2]. The sinking and equatorward flow of AABW is balanced by upwelling and poleward flow of Circumpolar Deep Water (CDW), forming the lower cell of the global overturning circulation[3]. In recent decades AABW has warmed[4], freshened[1,4] and decreased in volume and density[5], contributing to the increase in global ocean heat content and sea level rise[4,5]. The largest salinity and density trends have been observed in the Pacific and Australian Antarctic Basins, which are supplied by Ross Sea Bottom Water (RSBW)[4–9].

High salinity shelf water (HSSW) produced on the continental shelf of the Ross Sea is a precursor for RSBW. HSSW is formed on the continental shelf in winter by cooling and brine released during sea ice formation. HSSW exported from the continental shelf mixes with CDW as it descends the continental slope, producing AABW[10]. In addition, HSSW production in the western Ross Sea mediates the oceanic uptake of atmospheric $CO_2$ via both the solubility and biological pumps[11].

Freshening of RSBW has been linked to a decrease in salinity of HSSW in the Ross Sea, which freshened by $0.03\,dec^{-1}$ between 1958 and 2008 on the inner continental shelf[6,12]. A decrease in HSSW salinity of $0.05\,dec^{-1}$ was observed in Terra Nova Bay (TNB) between 1995 and 2006[13] and near the continental shelf break in the western (1995–2006) and central Ross Sea (1998–2006)[10].

Here, we use updated time series from five sites on the Ross Sea continental shelf to quantify salinity changes of HSSW between 1995 and 2018. We find the multi-decadal freshening identified in earlier studies persisted until 2014 and was then followed by a rapid increase in HSSW salinity to values previously observed in the 1990s.

## Results

**Study region**. We use hydrographic profiles collected during summer (December to February) to quantify salinity changes of HSSW in the Ross Sea. We focus on five areas, two where HSSW is formed in persistent polynyas (TNB and Ross Island (RI)) and three troughs through which HSSW is exported to the deep ocean (Drygalski Trough (DT), Joides Trough (JT) and Glomar Challenger Trough (GCT)) (Fig. 1). See Methods for details of the sampling in each region.

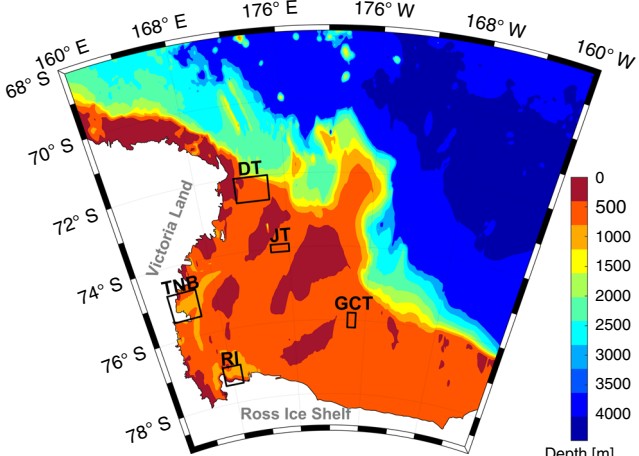

**Fig. 1** Study area in the Ross Sea. Bottom topography (m) is shown in colour. The five study areas are indicated with rectangles: Terra Nova Bay (TNB), Drygalski Trough mouth (DT), Ross Island depression (RI), Joides Trough (JT) and Glomar Challenger Trough (GCT). See Supplementary Fig. 1 for the location of the oceanographic profiles used.

**HSSW salinity variations from 1995 to 2018**. The mean salinity of the HSSW layer near the sea floor at each location is shown in Fig. 2. The most relevant time series for HSSW is TNB, where the saltiest and densest HSSW is found[14,15]. DT, through which the dense HSSW is exported to the continental slope[10,14,15], is also highly relevant, but closer to the shelf break and therefore more influenced by factors unrelated to HSSW formation. Between 1995 and 2014, salinity decreased at a similar rate at both locations: $-0.045 \pm 0.016\,dec^{-1}$ at TNB (significant at 99%) and $-0.043 \pm 0.035\,dec^{-1}$ at DT (significant at 98%) (where the error represents 95% confidence limits on the trends; see Methods for further information on the trends and their significance). The salinity decrease at TNB from 34.863 in 1995 to 34.763 in 2014 corresponds to a change in neutral density from 28.796 to $28.718\,kg\,m^{-3}$. A significant decrease in salinity was also observed at JT ($-0.037 \pm 0.015\,dec^{-1}$, significant at 99%). HSSW salinity decreased at RI ($-0.047 \pm 0.014\,dec^{-1}$, 1998–2014) and GCT ($-0.023 \pm 0.026\,dec^{-1}$, 1995–2008), but those records are short and incomplete and the trends are not statistically significant. These freshening trends are similar in magnitude to trends found prior to 2006 in previous work[10,13] and to those observed on the inner continental shelf between 1958 and 2008 ($-0.03\,dec^{-1}$)[6,12].

However, the updated time series reveal a sharp increase in salinity after 2014 at each location (except GCT, where the last observation was in 2008). By 2018, salinity in TNB had rebounded to values last observed in the mid-late 1990s (a salinity of 34.849, corresponding to a neutral density of $28.786\,kg\,m^{-3}$, Fig. 2). The recent salinity increase is large and rapid in comparison to salinity trends and variability observed prior to 2014: for example, the 2018 salinity anomaly at TNB was more than five standard deviations above the value expected if the pre-2014 trend had continued to 2018 (where the standard deviation is calculated from the de-trended pre-2014 record). The general freshening over the first 20 years of the record, and the sharp reversal after 2014, is seen throughout the water column at each location (Fig. 3).

The time series in Figs. 2 and 3 shows interannual variability superimposed on the overall salinity decrease between 1995 and 2014. Coherent fluctuations with a time-scale of 5–10 years are observed at each of the sites. Previous studies[16,17] found that anomalies in sea ice extent and export on similar time-scales were associated with variations in wind forcing, suggesting that wind-driven variability in sea ice export may have contributed to the interannual fluctuations in HSSW salinity. The increase in salinity after 2014 may be the most recent expression of the interannual variability evident in the HSSW salinity record, but the time series is too short to draw firm conclusions.

## Discussion

The salt content of the HSSW layer can change due to local or remote processes. The most likely local factor is variation in activity of the TNB polynya, where the most saline HSSW is found. However, sea ice production in the TNB polynya has changed little with time[18] and variations in the katabatic winds that drive polynya activity[19–22] show no correlation with the salinity time series (see Supplementary Discussion and Supplementary Fig. 5).

Moreover, near-synchronous variability at five locations on the continental shelf suggests that the observed salinity variability is driven by non-local processes. The salinity of HSSW likely reflects preconditioning by processes acting upstream, including sea ice formation and advection of freshwater from the east. The importance of preconditioning is further supported by the observation that salinity anomalies at each site extend throughout the water column (Fig. 3), with fresher HSSW associated with fresher upper ocean waters and a deeper halocline.

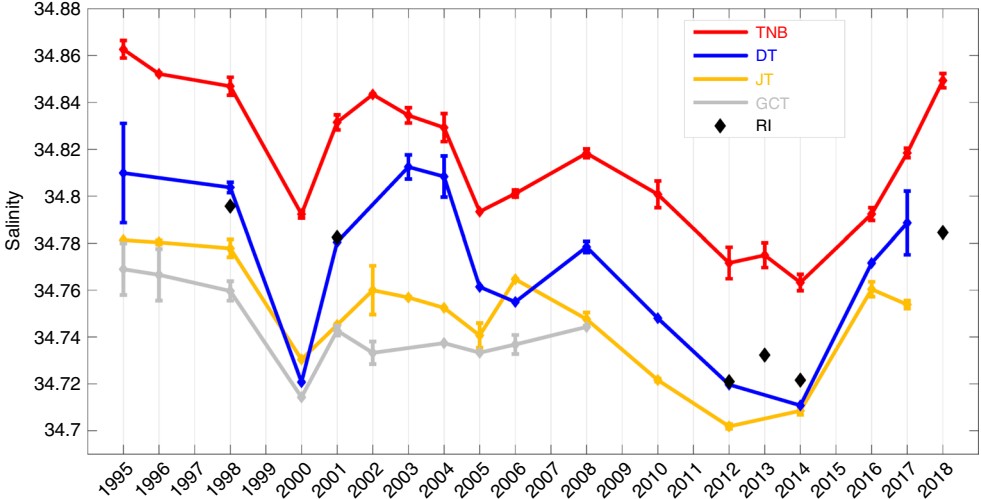

**Fig. 2** HSSW salinity time series (1995–2018) in the Ross Sea. Salinity averaged in the HSSW between 870 and 900 dbar in TNB (red line), between 850 and 880 dbar at RI (black diamonds), and in the deepest 20 dbar of the water column at DT (blue line), JT (amber line) and GCT (grey line). In each region, we have averaged CTD profiles on pressure surfaces to obtain a mean profile for each austral summer. The error bar is the mean standard deviation among all stations in the layer considered (see Methods) and is set equal to 0 in cases when only one profile was available in that year and region (see Supplementary Fig. 1 and Supplementary Table 1 for additional information on the number and location of CTD profiles used in each austral summer average).

Earlier studies have shown that increased continental ice discharge and melting in the Amundsen Sea supplied more than enough additional freshwater to account for the freshening of Ross Sea HSSW by $0.03\,\mathrm{dec}^{-1}$ between 1958 and 2008[6,12]. Freshening of the coastal and slope currents flowing from the Amundsen to the Ross Sea and oxygen isotope data provided further support to the hypothesis that increased freshwater input from the east could explain the freshening of Ross Sea shelf waters[6,12,23]. Float observations of freshening of the summer mixed layer[24] and model simulations of meltwater spread[25] also support the inference that freshwater from the Amundsen Sea influences the properties on the Ross Sea continental shelf. Other processes, including changes in the transport of CDW onto the continental shelf[26], precipitation, and melt of the Ross Ice Shelf were found to make smaller contributions to trends and variability of shelf water salinity[6,12,24].

In the absence of quantitative estimates of each of the terms contributing to the salt budget of the Ross Sea, and their variation in time, it is not possible to make a definitive statement regarding the causes of the multi-decadal freshening trend, or the rebound in salinity observed after 2014. However, a rough estimate of the change in salt or freshwater input needed to account for the salinity increase between 2014 and 2018 can be used to assess possible drivers. The salinity increase requires an addition of $6.322 \times 10^{15}\,\mathrm{kg}$ of salt (see Methods). If the salinity change reflected only a change in sea ice formation, the observed increase in salt content would require formation of an additional $255\,\mathrm{km}^3$ of sea ice, or an average annual anomaly of about $64\,\mathrm{km}^3\,\mathrm{yr}^{-1}$ (see Methods). For comparison the mean (1992–2013) cumulative sea–ice production of the Ross and TNB polynyas combined[18] is $438 \pm 64\,\mathrm{km}^3\,\mathrm{yr}^{-1}$. The increase in salinity of HSSW between 2014 and 2018 could therefore be accounted for by an increase in annual sea ice formation equivalent to 15% of the 1992–2013 mean ice production by the two polynyas, sustained over 4 years. (This value is likely an underestimate of the sea ice production anomaly required, as not all of the brine would accumulate in the HSSW).

A decrease in freshwater input to the HSSW layer could also contribute to the observed increase in salinity. Assuming the salt content of the HSSW remains unchanged, a reduction in volume of $0.018 \times 10^4\,\mathrm{km}^3$ would be needed to explain the observed increase in salinity (see Methods). This corresponds to a reduction in freshwater input of $180\,\mathrm{km}^3$ between 2014 and 2018, or about $45\,\mathrm{Gt}\,\mathrm{yr}^{-1}$ sustained over four years. Basal melt of ice shelves in the Amundsen Sea has been shown to vary on interannual and decadal timescales[27–29] and discharge of continental ice across the grounding line also varies from decade to decade, with anomalies of the same order of magnitude as the required change in freshwater input[30]. However, mass loss from the West Antarctic Ice Sheet has increased in each of the past three pentads (from $-65 \pm 27\,\mathrm{Gt}\,\mathrm{yr}^{-1}$ in 2002–2007, to $-148 \pm 27\,\mathrm{Gt}\,\mathrm{yr}^{-1}$ in 2007–2012, and $-159 \pm 26\,\mathrm{Gt}\,\mathrm{yr}^{-1}$ in 2012–2017)[31], which would act to decrease rather than increase the salinity of shelf waters in the Ross Sea.

The anomalies in sea ice formation or meltwater input needed to account for the observed increase in salinity of HSSW between 2014 and 2018 are large, relative to their mean values and variability, but of the right order of magnitude. These rough calculations therefore suggest that an increase in sea ice formation and/or a reduction in freshwater input could explain the recent increase in HSSW salinity. The salinity increase is unusual in the observational record and requires a climate anomaly of sufficient magnitude to reverse 20 years of freshening at the multi-decadal trend observed prior to 2014. Further work is needed to identify the physical mechanisms responsible for the salinity increase and their link to larger-scale climate phenomena. Contributions to the salt budget of the Ross Sea must be better observed and understood, in particular formation and export of sea ice and inflow of freshwater from the Amundsen Sea.

RSBW is formed from a mixture of HSSW and CDW. Just as the multi-decadal freshening of HSSW caused a reduction in the salinity and density of RSBW[4–7], the recent shift to saltier HSSW will result in a rebound in the salinity and density of RSBW, if changes in CDW or mixing do not compensate for the increase in HSSW salinity. If the shift to saltier HSSW and RSBW persists, this will have repercussions for abyssal stratification and ventilation, oceanic $CO_2$ sequestration, ocean heat storage, and the lower limb of the global overturning circulation.

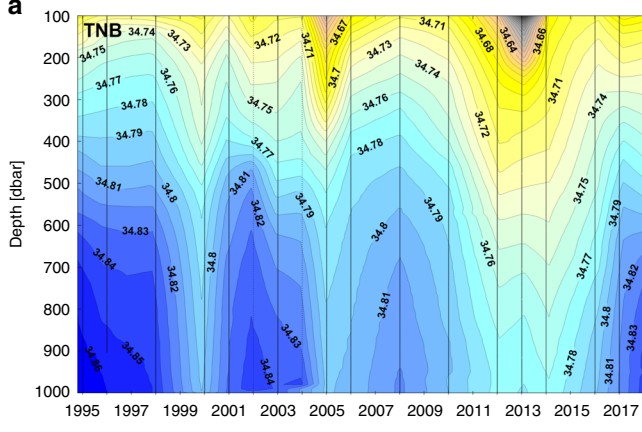

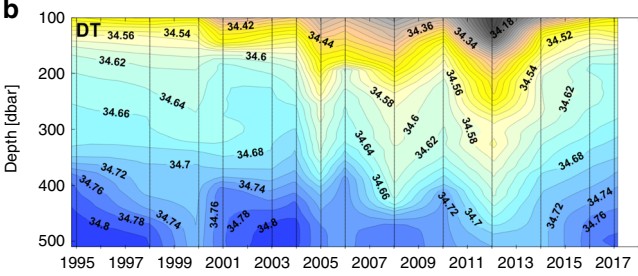

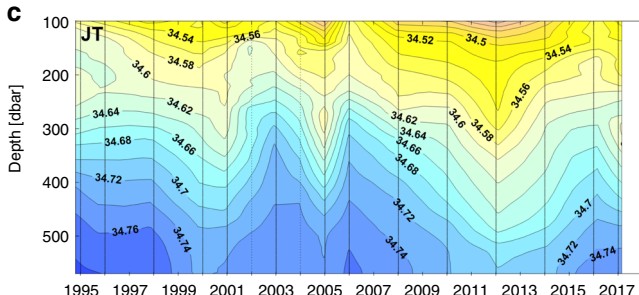

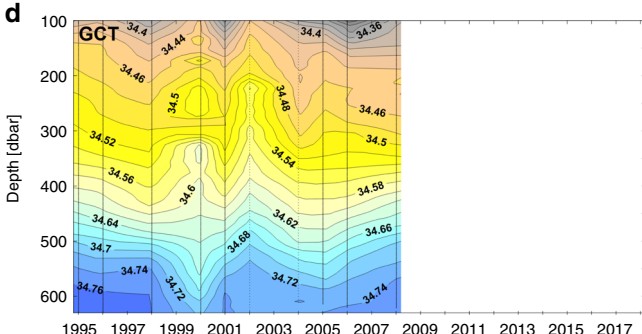

**Fig. 3** Salinity as a function of time and depth at each study site. Time-depth isopleths of salinity over the study areas of **a** Terra Nova Bay (TNB), **b** Drygalski Trough mouth (DT), **c** Joides Trough (JT) and **d** Glomar Challenger Trough (GCT). The vertical black lines indicate sampling years. In each region, we have averaged CTD profiles on pressure surfaces to obtain a mean profile for each austral summer.

## Methods

**Oceanographic observations.** The hydrographic measurements were obtained during 16 Italian National Antarctic Research Programme (PNRA) expeditions from 1995 to 2017 and three Nathaniel B. Palmer expeditions in 2004, 2013 and 2018 respectively as part of the ANSLOPE, TRACERS and CICLOPS projects. The CTD data were obtained using a Sea-Bird Electronics SBE 9/ 11+. The CTD was equipped with dual temperature-conductivity sensors flushed by a pump at a constant rate. Calibrations were performed before and after the cruises. Data were acquired at the maximum frequency (24 Hz). Hydrographic data were corrected

and processed according to international procedures[32]. We used a total of 221 CTD casts: 124 in TNB, 28 in DT, 39 in JT, 22 in GCT and 8 in the RI region. The locations of all the casts in each region are shown in Supplementary Fig. 1 and details of the sampling in each year are summarised in Supplementary Table 1. In each region (Fig. 1 and Supplementary Fig. 1), we averaged profiles on pressure surfaces of 1 decibar to obtain a mean profile for each region. In TNB, we used profiles collected in the area between 74.75°S–75.50°S and 163.00°E–166.00°E with station depths deeper than 800 m, thus we have selected only the stations in the Drygalski Trough where the HSSW accumulates. At the DT mouth we have chosen the region between 72.00°S and 72.67°S and 171.50°E and 174.50°E, with casts deeper than 500 m and shallower than 530 m, in order to use only the profiles that intercept the outflow of HSSW near the bottom of the trough. In JT and GCT, the stations deeper than 500 and 600 m, respectively have been selected to capture the HSSW outflow. Profiles were averaged between 73.90°S–74.10°S and 174.20°E–176.00°E for JT, and between 75.80°S–76.20°S and 178.00°W–177.10°W for GCT. At RI we selected stations deeper than 800 m in the region 77.00°S–77.50°S; 167.00°E –169.20°E, in order to sample the HSSW that accumulates in the depression.

**Salinity time series and error estimates.** The time series in Fig. 2 were obtained by averaging the salinity in the 30 dbar layer from 870 to 900 dbar for TNB, in the layer from 850 to 880 dbar for RI, and in the bottom 20 dbar for DT, JT and GCT. For DT, JT and GCT we averaged the bottom 20 dbar in order to capture only the Shelf Water outflow (the presence or absence of MCDW may contribute to salinity changes at shallower depths). The error bar in Fig. 2 is the horizontal standard deviation among available stations in each region and each year, evaluated on each 1 dbar pressure surface, and averaged over the layer considered (see Supplementary Fig. 2). When only 1 profile was available in a particular year and region the error bar is set equal to 0. Supplementary Fig. 2 shows that despite the sparse spatial distribution of the CTD casts within the TNB region, the horizontal (spatial) standard deviation is low, especially in the bottom layer that is the main focus of our work. The small spatial standard deviation indicates that differences in station locations from year-to-year are unlikely to alias the results.

**Potential sampling biases.** We use only summer data, to avoid seasonal sampling biases. However, the dates of the observations vary from year to year and between regions. To assess the potential for sampling bias related to temporal variations, we examined time series of salinity measured by a mooring of the Italian Marine Observatory in the Ross Sea (MORSea) located in the middle of TNB (75.14°S, 164.55°E). We calculated the standard deviation for the time period over which the CTD samples were obtained. Most of the CTD profiles in TNB were collected during January and February, but during summer 1997/1998 the first CTD was made on December 7 and in 2013 the last CTD was collected on March 5. Therefore, we calculated the standard deviation for the time period from December 7 to March 5 for each moored time series (Supplementary Fig. 3). The standard deviation is about 0.002 at 836, 823 and 973 m during the austral summers of 1996/1997, 2000/2001 and 2006/2007, respectively. At greater depth (1076 m) in 2014/2015 the standard deviation was 0.001. The standard deviation of each of the moored times series is small compared to the interannual variations we observe in the time series in Fig. 2. Furthermore, Supplementary Fig. 3 shows that during the CTD sampling period (December to March), the salinity measured by the mooring is close to the maximum values reached during the year. The small spatial and temporal standard deviations compared to the year-to-year changes in salinity confirms that the results in Fig. 2 are not the result of sampling biases.

For regions outside of TNB, motion of fronts or water mass property gradients could introduce salinity variability unrelated to HSSW variability, especially for regions close to the continental shelf break. We therefore selected only CTD profiles that capture the outflow of the dense HSSW. We used stations located in the middle of the troughs and profiles that sampled HSSW in the bottom layer, where HSSW is defined[15] as water with neutral density ($\gamma^n$) > 28.27 kg m$^{-3}$ and potential temperature $\theta$ < −1.85 °C. As found in TNB, the spatial (horizontal) standard deviation for the different regions is small compared to the salinity changes between years. The sampling time period spans from December 26 to February 26 at DT; from December 16 to February 15 at JT; from December 13 to February 12 at GCT and from January 23 and February 12 at RI.

**Linear trends.** We estimated the linear trend for each of the time series shown in Fig. 2 (Supplementary Fig. 4) and determined the 95% confidence intervals, the coefficient of determination ($R^2$), and the consistency of the trends through the Mann–Kendall test[33,34]. The linear trends from 1995 to 2014 are statistically significant at the 99% (TNB and JCT) or 98% level (DT). The short and incomplete records at RI and GCT also show negative trends, but the trends are not statistically significant.

**Salt budget calculations.** The addition of salt required to account for the salinity increase in HSSW between 2014 and 2018 was calculated as follows. The increase in salt content of the HSSW layer is $\rho V \Delta S = 6.322 \times 10^{15}$ kg of salt, where $\rho = 1027$ kg m$^{-3}$ is the density of sea water, $\Delta S = 0.086$ is the change in salinity, and $V = 7.158 \times 10^4$ km$^3$ is the mean volume of HSSW on the Ross Sea continental shelf[15].

The mass of salt added to the water column during sea ice formation is $f\rho_{ice}V_{ice}S_{surf}$, where $f$ is the fraction of salt released during freezing of sea ice (0.79)[35], $\rho_{ice}$ is the density of sea ice (920 kg m$^3$), $V_{ice}$ is the volume of sea ice formed and $S_{surf} = 34.0$ is the salinity of surface water. If sea ice formation was solely responsible for the increase in salinity, the observed increase in salt content would require formation of an additional 255 km$^3$ of sea ice, or an average annual anomaly of about 64 km$^3$ yr$^{-1}$.

The increase in HSSW salinity could also reflect a reduction in freshwater input to the HSSW layer. Assuming the salt content of the HSSW remains unchanged ($\rho_o V_o S_o = \rho_o V_f S_f$, where the subscripts refer to initial and final values), a reduction in volume of $0.018 \times 10^4$ km$^3$ would be needed to explain the observed increase in salinity.

## Data availability

The CTD data obtained during 16 Italian National Antarctic Research Programme (PNRA) expeditions from 1995 to 2017, that support the findings of this study are available from the corresponding author upon request.

The 2004 CTD data (NODC Accession number: 0036202) that support the findings of this study are available from the Lamont-Doherty Earth Observatory at http://ocp.ldeo.columbia.edu/res/div/ocp/projects/anslope/Data.html. The 2004 CTD data are also available from the National Centers for Environmental Information (NCEI) repository at https://accession.nodc.noaa.gov/0036202. Mele, Philip A.; Columbia University, Lamont-Doherty Earth Observatory (2011). Oceanographic temperature, salinity, dissolved oxygen, and pressure measurements collected using CTD from Nathaniel B. Palmer in the Ross Sea during 2004 (NCEI Accession 0036202). NOAA National Centers for Environmental Information. Dataset. https://accession.nodc.noaa.gov/0036202.

The 2013 CTD data that support the findings of this study have been deposited in the Marine Geoscience Data System (MGDS) repository with the identifier data DOI code "10.1594/IEDA/320068", at http://www.marine-geo.org/tools/search/Files.php?tab=datacitation&data_set_uid=20068. Hansell, D. (2015). Calibrated Hydrographic Data from the Southern Ocean acquired with a CTD during the Nathaniel B. Palmer expedition NBP1302 (2013). Interdisciplinary Earth Data Alliance (IEDA). https://doi.org/10.1594/IEDA/320068.

The 2018 CTD data that support the findings of this study are available from CICLOPS project P.I. and contributing author of this study Giacomo R. DiTullio upon request. After March 1, 2020, the data will be available from the NSF BCO-DMO database website: https://www.bco-dmo.org/project/774945.

The time series of salinity measured by the mooring of the Italian MORSea that support the findings of this study are available from the corresponding author upon request.

The in situ meteorological data collected in Terra Nova Bay by the Automatic Weather Stations (AWS) Rita, between 1995 and 2017, that support the findings of this study (Supplementary Discussions) are available from the MeteoClimatological Observatory at MZS and Victoria Land of PNRA at http://www.climantartide.it.

## Code availability

The Matlab scripts used to analyse the data and to generate the Figures in this paper are available from the corresponding author on request.

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

## Acknowledgements

Logistical and financial supports for these study was provided by the Italian National Programme for Antarctic Research (PNRA), grants: 2016/AN2.02 (CELEBeR); 2016/A3.06 (P-ROSE); OSS-13 (MORSea); 2009/A2.04 (T-Rex); 2004/08.03 (CLIMA IV). This study was partially supported by a grant from the US National Science Foundation to G.R.D. (OPP-1644073). S.R.R. was supported by the Centre for Southern Hemisphere Oceans Research, a collaboration between the Qingdao National Laboratory for Marine Science and Technology (QNLM) and CSIRO; by the Australian Antarctic Programme Partnership; and by the Earth Systems and Climate Change Hub of the Australian Government's National Environmental Science Programme. We are thankful to the Meteo-Climatological Observatory' of the PNRA for the meteorological data set. We are

also thankful to CoNISMa for the management of P-ROSE project. We thank the CLIMA IV, T-Rex, MORSea, CELEBeR and P-ROSE teams and the crew of the RV Italica who provided excellent support during the field operations.

## Author contributions

P.C., P.F., G.F. and G.B. conceived the study. P.C. directed this work. P.C., G.R.D., P.F., G.F., G.S. and G.B. contributed to oceanographic data collection. P.C. assembled and analysed the oceanographic observational data. V.C. analysed the in situ meteorological data. P.C., P.F., S.R.R. and G.B. contributed to the interpretation of the oceanographic data and to the interpretation of the results. V.C. and G.F. contributed to the interpretation of the in situ meteorological data and to the interpretation of the results. P.C. and S.R.R. wrote the paper. P.F. contributed to writing the paper. All authors discussed the results, commented and contributed to the final version of the paper. G.B. supervised the study.

## Competing interests

The authors declare no competing interests.
