## [Peer Review File · Nature Communications]

Reviewers' comments:

Reviewer #1 (Remarks to the Author):

This manuscript describes the results from a unique set of hydrographic observations that taken together comprise the basis for a time series (1995-2018) describing temporal variations in Ross Sea High Salinity Shelf Waters (HSSW) from their formation sites in the south to where they exit the basin to the north. The main result, a freshening over the last 3-4 years of HSSW on the heels of a number of years of the salinification is interesting, exciting and important. Interesting and exciting as freshening of Antarctic Bottom Waters has been reported from multiple locations throughout the Southern Ocean and important, because when combined with the almost ubiquitous warming suggests a possible mechanism for deceleration of the abyssal contribution to sea level rise.

The supplementary material describes in fairly simple terms a comparison of the salinity time series at five different locations in the Ross Sea to major local weather event statistics and the authors conclude that forcing supporting neither the interannual nor the longer term variability in the salinity signal is likely to be locally driven. I agree with the conclusions. The manuscript is well written and is suited to the journal. As I suspect that most of what I ask for has already been done, or least would not take long to do, my recommendation is to accept subject to revision.

That being said I have two major concerns and a disappointment. These are listed below (a-c) followed by specific comments, suggestions and critique, some of which support the major concerns (a-c), others are minor concerns, and still others are trivia.

a. Uncertainties: Although error bars are provided on Figure 2 and the supplementary materials provide the depth ranges and areas over which the 1-decibar samples were taken, there is no indication as to how many casts went into the averages shown. Values in the vertical are expected to be highly correlated, especially when chosen to represent a single water mass. Therefore, to interpret the error bars it is important to know whether they are based on a single cast or multiple casts as well as the timing and length of the period over which the samples were obtained (once some time during the season or multiple times through out). Even more importantly, the estimates for the trends had no uncertainties attached to them at all. Compounding the problem, the trend estimates were compared to and found to be "consistent" with estimates from the literature as recent as 2011 that also had no associated uncertainties. Without estimated uncertainties consistency cannot be determined.

b. The framing on the result: First, I really like the title that uses the word "rebound" to describe the result. It seems clear to me, though I am willing to listen to why the authors may feel I have misinterpreted the result, that the recent freshening is the part of appears to be either a 4-6 year or multi-decadal cycle in HSSW salinity. Yet, it framed it terms of what the authors describe as a long-term 1995 to 2014 trend. Please see (10) below. Also, I would like to see the trend quantified, because I suspect that when uncertainties are attached, the strong interannual variations will render

it not significantly different from zero in at least some of the locations. In particular, in the DT and GCT. In the RI, the result is probably significant but only because of the lack of data points. To me the freshening since 2014 is definitely an important result, but it should be framed in terms of strong 4-6 year variations, as well as the hint of a much longer term signal that this time series is not yet long enough to capture.

c. I recognize that the focus here is on the salinity signal, however in these waters, salinity dominates density and therefore it would have been interesting to have seen either some interpretation in density space or discussion about the effects on density. And please see my comment on Figure 3 below.

Specific Critique (L represent Line):

1. L19-20: From the start, the statement “HSSW salinity rebounded sharply after 2014, with values in 2018 similar to those observed in the mid-1990s” begs the question, why the comparison to mid-1990s when there were also high values in the early 2000s (TNB, DT), late 1990s (RI) and mid 2000s (JT). Agreed, these details should not necessarily be mentioned here, but their existence highlights the variability as opposed to the trend.
2. L31-32: “is the primary source of oxygen to the abyss” needs a reference
3. L37: Could include Menezes et al., 2017 in this list of citations.
4. L39: TNB either needs to be defined here or not used until the next paragraph
5. L42: Point the reader to the supplementary material describing the data and indicate either here or in Section S1 whether this is an extension of the earlier studies – i.e. is the data record prior to 2006, the same as used in Budillion et al, 2011 etc.
6. L56-58: Please include estimates of uncertainty on these trends.
7. L58: The word “consistent” cannot be used as neither this work nor the previous literature supply estimates of uncertainties.
8. L60: Say something about the trends or lack thereof in the GCT and RI.
9. L69-70: ...the updated time series now reveal a sharp increase in salinity at each location all locations where data are available after 2014.
10. L74-75: Agreed “rebound” or “reversal” (though not quite the same thing) in 2014. But, it’s hard to see trends in Figure 3 and even in Figure 2 the trends seem highly dependent upon where one begins the time series. E.g. if the time series began in 2000, there would be no apparent trends. So if it began in 1990 - would we see trends? My take is that indeed when the available data only spanned the time period before 2014 the interpretation was strong interannual variability superimposed on an overall salinity decrease, but with the extended record this now looks like interannual variability with no obvious (significant?) downward trend. For now, the upswing beginning in 2014 looks like that, which occurred in 2000/2001 and 2005/2006, and certainly begs the question - what will happen next?

11. L76: This quoted lag only seems to apply to the time series prior to 2008. Was a correlation analysis done to obtain estimates of lag?
12. L96: and this same reference suggests that there were large ice reduction events in the Ross Ice Shelf Polynya in 2000 and 2002.
13. L123-124: So here, I see an opportunity with this extended data set to point out that what was seen earlier as a long term trend, may not be that at all, but could be the downward swing of a multi-decadal oscillation. Continued observation and modeling will eventually provide further insight.
14. L156: And on that note, I would include here the need for continued observation of these regions and maybe even improved time series in these particular regions. The latter, is hard to judge given the minimal amount of information provided about the hydrographic data set used.

Figures:

Figure 1: The color scale is such that although the shallower areas are orange, everywhere else of interest is red and invisible. Perhaps include a few contour lines or change the color scale so that the defining bathymetry of the troughs can be seen.

Figure 2: Suggest including the grid in both directions to enhance comparisons of salinities. Axes tick labels are too small.

A figure or subpanel showing the quantified trends for this analysis including estimates of uncertainties would substantially improve the grounds for the statements made in the text. I played around with numbers that I pulled from the manuscript figure 2 and got the results shown in the attached figure.

Figure 3: Is it sensible to average in pressure space? I would like to see a figure or even just statement indicating the flatness of the isopycnals in the regions of interest.

Figure S1: To recognize the similarities and differences in these time series it would be useful to have these plots either distributed vertically, or better yet, all in the same figure (yes, the latter would be more complicated as it would require multiple axes).

Also, what are the definitions of severity and intensity? It is interesting that they are so different. Also, same as above, are the trends significant given the strong variability?

Reviewer #2 (Remarks to the Author):

Dense water export from the Ross Sea is an important issue. The fact that salinity has been increasing since 2014/2015 and is represented by observations from more than one year is important.

There should be some indication somewhere in the paper of the times when data were collected. The missing symbols in Fig 2 tell the story, but it would be nice to have a simple table somewhere indicating which years have observations from each of the regions, maybe give the month of each year.

When during the year were these observations made? I assume sometime during Nov to Feb of the indicated year. Given the page constraints, maybe a simple statement that most measurements were in early? mid? late? summer with the explicit times in the supplemental information? This timing during the year will only affect the surface salinity in any case.

I don't agree that surface processes don't affect deep salinity (lines 104-106). Deep convection in winter over most of the Ross Sea takes surface conditions throughout the water column. So current deep water salinity can certainly be affected by surface salinity in the previous summer as well as import of water from the Amundsen shelf (although with a longer delay than one year).

The conclusions of the paper are a bit nebulous: might be winds, might be ice export, might be Amundsen Sea import, might be CDW import, etc. I think there is enough room in this paper for a few plausibility estimates. How much would sea ice export need to change to produce the indicated salinity increase? How much would meltwater in the Amundsen need to decrease to reduce meltwater import sufficiently to produce the indicated change? Would more CDW import make a difference? Combination of these effects? These estimates might add a paragraph to the paper and at least point readers to likely processes.

I am confused about how the contours in Fig 3 were created. There seem to be extrema in salinity during some years when there are no observations. There is also a bit of structure in the contours between observations. The authors need to explain how this is possible or what additional information is used to create these structures.

Fig 2: There should be different color choices for TNB and JT curves. Although the higher values should clearly be in Terra Nova Bay, so with a bit of thought the reader should be able to tell.

Reviewer #3 (Remarks to the Author):

The authors identify a rapid rebound of salinity in the Ross Sea, after a previously well-documented minimum was reached in the period 2012-2014. Identifying changing salinity in this region is important as the salinity of HSSW helps determine the production rate and water mass characteristics of Ross Sea Bottom Water (RSBW), a major contributor to Antarctic Bottom Water (AABW) that is the largest abyssal water mass in the global ocean.

Assuming that the results are robust (see Major Comments, below), the observational result justifies publication in Nature Comms., with the potential to be motivational to further research and, therefore, be well cited. The paper is short, and could even be shortened further since the paragraph devoted to Conclusions (lines 138-158) basically restates material that has just been reported in the previous paragraphs. However, the analysis of causality to explain the observation requires much more care: also see Major Comments, below.

MAJOR COMMENTS

1. Much more information is needed on sampling per region per year: Information provided about the CTD data is totally inadequate for assessing whether the results might have arisen from sampling biases. How many CTD profiles are included in each region in each sampled year? Is the spatial distribution of each set consistent with the assumption that mean profile values can be reasonably compared? Given that interannual (and annual) variations are large in Ross Sea hydrographic conditions, might apparent changes in hydrography for a specific region be caused by motion of fronts or water mass property gradients, and/or a different “centroid” of sampling? The authors might argue that the error bars on Figure 1 “answer” these questions, but they don’t (for me) and in any case they not explained.

2. The authors conclude that changes in freshwater input from the Amundsen Sea explains the changes, but this is mainly based on (i) observed changes in Amundsen Sea ice sheet mass loss rates from Konrad et al. 2017 and a couple of papers looking at single ice shelves and grounded-ice catchments and (ii) rejecting one specific hypothesis – that variability of some aspect of katabatic winds in Terra Nova Bay (TNB) could drive changes in dense water production – which they reject on the basis of Figure S1 that TNB katabatics are *not* responsible, even though they don’t present a specific mechanistic link between individual katabatic metrics and salinity. I believe the choice of

hypothesis to reject is based on the availability of a katabatics record from an AWS, not on being confident that it is the only alternative to the Amundsen Sea advection freshwater source.

My intuition says that the authors are correct, but then why put so much effort into one specific negative hypothesis (katabatics in TNB) rather than either also dismissing other hypotheses (Ross Shelf Polynya variability? Freshwater from Ross Ice Shelf? Residual effects of the 2001-2005 iceberg “blockage” around Ross Island, given retention time scales on the continental shelf? ???) or doing the best possible analyses for the “positive” hypothesis of Amundsen Sea freshwater? For example:

a) Show that the change in Ross Sea salinity is consistent with the magnitude of variability in the freshwater equivalent of the Amundsen Sea ice sheet mass balance (and/or sea ice advected westward);

b) Look at sea ice motion vectors as a guide to changes in annual flux of Amundsen Sea waters into the eastern, and subsequently all of the Ross Sea.

c) Look at the weather patterns (from reanalyses like ERA-Int etc.) to seek evidence that ocean flows from the Amundsen to the Ross should have decreased in the last few years.

d) Create (and show) sea ice metrics that make sense in view of proposed mechanisms. You state that sea ice is acting the wrong way to explain the recent trend in salinity, but perhaps there is an EOF or other analysis that pops out the signal you want to see. There are recent papers, some of which you cite (e.g., cites 28-31) that discuss some of the “external” mechanisms responsible for recent rebounds in Antarctic climate. These should probably be re-read and used to provide a more robust defense of the mechanistic relationship proposed between the Amundsen Sea and the Ross salinity.

3. Lines 103-106. You comment that “local processes affecting the surface layer” cannot explain freshening below the surface layer. This is not true. If diapycnal mixing is important, then the fresh signal propagates downward. Dumping more fresh water into the SML in summer also reduces the salinity to which subsequent winter convection might drive full-depth averaged salinity (e.g., in the central Ross Sea, there is interannual variability in the depth of the winter mixed layer.)

Furthermore, there are feedbacks between circulation and stratification so that, while a fresher cap might prevent the deeper water from mixing with the SML in a 1D sense, there might be a change in the 3D flow field that causes correlated changes in deep hydrography.

MINOR COMMENTS

Lines 38, 39, maybe elsewhere: You can't talk of "freshened by" then give a negative number; same for "decrease in salinity".

Figure 2: Colors for TNB and JT are too close together. I could only sort them out by knowing that TNB salinity should be higher.

Lines 114-115: => "are believed to make smaller contributions to".

Lines 138-158: Given how short the paper is, even though it would be longer if you follow by Major Comments, a simple restatement of what has already been said doesn't seem worthwhile. Maybe expand upon the "consequences" paragraph, which is now a short final paragraph here.

Given the presence of a large ice shelf, be pedantic and always say either "CONTINENTAL shelf" or "ICE shelf" when you refer to "shelf".

-- Laurie Padman

Reviewers' comments:

In black the Reviewer comments, in red our responses.

For all the reviewers:

We thank the reviewers for their insightful comments. The manuscript has been heavily revised, with additional information on the sampling, estimates of uncertainty, and the mechanisms potentially responsible for the recent increase in salinity. In addition, the time series for the Drygalski Trough mouth (DT) and Joides Trough (JT) have been revised. The previous plots used criteria that sometimes included modified Circumpolar Deep Water (mCDW) in the averages for each austral summer. We have refined the criteria used to ensure the time series presented represents temporal variations in the salinity of HSSW, so that each of the time series is consistent.

Reviewer #1 (Remarks to the Author):

This manuscript describes the results from a unique set of hydrographic observations that taken together comprise the basis for a time series (1995-2018) describing temporal variations in Ross Sea High Salinity Shelf Waters (HSSW) from their formation sites in the south to where they exit the basin to the north. The main result, a freshening over the last 3-4 years of HSSW on the heels of a number of years of the salinification is interesting, exciting and important. Interesting and exciting as freshening of Antarctic Bottom Waters has been reported from multiple locations throughout the Southern Ocean and important, because when combined with the almost ubiquitous warming suggests a possible mechanism for deceleration of the abyssal contribution to sea level rise.

The supplementary material describes in fairly simple terms a comparison of the salinity time series at five different locations in the Ross Sea to major local weather event statistics and the authors conclude that forcing supporting neither the interannual nor the longer term variability in the salinity signal is likely to be locally driven. I agree with the conclusions. The manuscript is well written and is suited to the journal. As I suspect that most of what I ask for has already been done, or least would not take long to do, my recommendation is to accept subject to revision.

That being said I have two major concerns and a disappointment. These are listed below (a-c) followed by specific comments, suggestions and critique, some of which support the major concerns (a-c), others are minor concerns, and still others are trivia.

- a. Uncertainties: Although error bars are provided on Figure 2 and the supplementary materials provide the depth ranges and areas over which the 1-decibar samples were taken, there is no indication as to how many casts went into the averages shown.

Values in the vertical are expected to be highly correlated, especially when chosen to represent a single water mass. Therefore, to interpret the error bars it is important to know whether they are based on a single cast or multiple casts as well as the timing and length of the period over which the samples were obtained (once some time during the season or multiple times throughout). Even more importantly, the estimates for the trends had no uncertainties attached to them at all. Compounding the problem, the trend estimates were compared to and found to be “consistent” with estimates from the literature as recent as 2011 that also had no associated uncertainties. Without estimated uncertainties consistency cannot be determined.

We have added a table (Table. S1) that includes additional detail on the sampling, including the number and dates of the profiles used for each austral summer average. In each region, changes in time are assessed from profiles that are close in space and from the same season. We have added a new paragraph in the supplementary section “S.2 Time series and Linear trends”, to address the possible bias related to the sampling.

The error bars are now given as the horizontal standard deviation of the profiles used to compute each austral summer average, evaluated on each 1-dbar pressure surface and averaged over the layer considered (see paragraph S2).

b. The framing on the result: First, I really like the title that uses the word “rebound” to describe the result. It seems clear to me, though I am willing to listen to why the authors may feel I have misinterpreted the result, that the recent freshening is the part of appears to be either a 4-6 year or multi-decadal cycle in HSSW salinity.

Yet, it framed it terms of what the authors describe as a long-term 1995 to 2014 trend. Please see (10) below. Also, I would like to see the trend quantified, because I suspect that when uncertainties are attached, the strong interannual variations will render it not significantly different from zero in at least some of the locations. In particular, in the DT and GCT. In the RI, the result is probably significant but only because of the lack of data points. To me the freshening since 2014 is definitely an important result, but it should framed in terms of strong 4-6 year variations, as well as the hint of a much longer term signal that this time series is not yet long enough to capture.

We assume the reviewer meant to say “recent salinification” rather than recent freshening (i.e. referring to the increase in salinity between 2014 and 2018). We agree it is possible that the recent increase in salinity is a more extreme expression of interannual variability seen earlier in the record (and now include a sentence to this effect at line 99-101). However, the time series is too short (capturing only about 2 “cycles”) to say much about the nature or timescale of any cycle. We note that the minima and maxima of the TNB time series, for example, are separated in time by between 5 and 10 years, so it is not clear that there is any cyclical behaviour with a well-defined period.

We also note that the rate, magnitude and duration of the recent “rebound” in salinity is large compared to interannual variability seen earlier in the record. If a similar climate forcing is responsible for all of the interannual variability observed in the full record (a

hypothesis that cannot be tested with available data), the forcing anomaly during the 2014-2018 period must have been unusually strong.

In addition, it is not true that the interannual variations render the pre-2014 trend insignificant. We have now provided further details with regard to the uncertainty in the trends. The pre-2014 trends are significant at the 98% level or better at TNB, DT and JT. The trends are also negative, but not significant, for the short and incomplete records at RI and GCT. As noted in the text, the pre-2014 freshening trend found here is consistent with a number of other studies of Ross Sea observations dating back to the 1950s (e.g. Jacobs and Giulivi, 2010).

c. I recognize that the focus here is on the salinity signal, however in these waters, salinity dominates density and therefore it would have been interesting to have seen either some interpretation in density space or discussion about the effects on density. And please see my comment on Figure 3 below.

As salinity dominates density in this region, the density time series look very similar to those of salinity. For this reason, we don't think it is very useful to show the density plots as well. However, we have included some neutral density values in the text (Line 82-84), so readers can assess what the changes in salinity mean for density.

Specific Critique (L represent Line):

1. L19-20: From the start, the statement "HSSW salinity rebounded sharply after 2014, with values in 2018 similar to those observed in the mid-1990s" begs the question, why the comparison to mid-1990s when there were also high values in the early 2000s (TNB, DT), late 1990s (RI) and mid 2000s (JT).

We have compared the 2018 salinity to the mid-late 1990s because the salinity is higher at that time than in the 2000s across each of the time series (with the exception of 2003 and 2004 at DT, when they are similar).

Agreed, these details should not necessarily be mentioned here, but their existence highlights the variability as opposed to the trend.

2. L31-32: "is the primary source of oxygen to the abyss" needs a reference

done

3. L37: Could include Menezes et al., 2017 in this list of citations.

done

4. L39: TNB either needs to be defined here or not used until the next paragraph

done

5. L42: Point the reader to the supplementary material describing the data and indicate either here or in Section S1 whether this is an extension of the earlier studies – i.e. is the data record prior to 2006, the same as used in Budillon et al, 2011 etc.

It is not an extension of the record used by Budillon et al, 2011. Here we use a different approach in selecting the CTD casts for the area, to minimize the potential effects of sampling bias.

6. L56-58: Please include estimates of uncertainty on these trends.

done

7. L58: The word “consistent” cannot be used as neither this work nor the previous literature supply estimates of uncertainties.

Changed the sentence to “These freshening trends are similar in magnitude with trends found prior to 2006 in previous work^{6,14} and to those observed on the inner continental shelf between 1958 and 2008 (-0.03 dec-1)^{4,5}”

8. L60: Say something about the trends or lack there of in the GCT and RI.

added

9. L69-70: ...the updated time series now reveal a sharp increase in salinity at each location all locations where data are available after 2014.

changed

10. L74-75: Agreed “rebound” or “reversal” (though not quite the same thing) in 2014. But, it’s hard to see trends in Figure 3 and even in Figure 2 the trends seem highly dependent upon where one begins the time series. E.g. if the time series began in 2000, there would be no apparent trends. So if it began in 1990 - would we see trends? My take is that indeed when the available data only spanned the time period before 2014 the interpretation was strong interannual variability superimposed on an overall salinity decrease, but with the extended record this now looks like interannual variability with no obvious (significant?) downward trend. For now, the upswing beginning in 2014 looks like that, which occurred in 2000/2001 and 2005/2006, and certainly begs the question - what will happen next?

Whether the recent increase in salinity represents a sustained shift in the system or a large but short-lived reversal of the long-term freshening trend is unknown at this stage. We agree that “what will happen next” is an interesting question, one that we will need to wait to answer.

11. L76: This quoted lag only seems to apply to the time series prior to 2008. Was a correlation analysis done to obtain estimates of lag?

The updated time series show near-synchronous changes at each of the five sites. The lag apparent in the earlier plots was an artefact of including some mCDW in the time series of HSSW properties.

12. L96: and this same reference suggests that there were large ice reduction events in the Ross Ice Shelf Polynya in 2000 and 2002.

We now mention that iceberg calving events have altered polynya activity and cite Robinson et al. (2012). However, as noted in the text, there is no relationship between the time series of HSSW salinity and changes in activity of the Ross Ice Shelf Polynya associated with iceberg calving. This may be because reductions in sea ice formation by the Ross Ice Shelf Polynya in those years were compensated by increases in sea ice formation outside of the polynya.

13. L123-124: So here, I see an opportunity with this extended data set to point out that what was seen earlier as a long term trend, may not be that at all, but could be the downward swing of a

multi-decadal oscillation. Continued observation and modeling will eventually provide further insight.

We agree with the reviewer that what has been interpreted as a long-term trend (extending back to 1958, according to Jacobs and Giulivi, 2010) may instead be part of a multi-decadal oscillation. But until we have a longer record, this is speculation. We only have four years of data showing an increase in salinity so we feel it is inappropriate here to speculate in this way.

14. L156: And on that note, I would include here the need for continued observation of these regions and maybe even improved time series in these particular regions. '

We now highlight in the concluding paragraphs the need for improved observations of the physical processes influencing the salt budget of the Ross Sea.

The latter, is hard to judge given the minimal amount of information provided about the hydrographic data set used

We have added more information on the dataset used, showing that sampling bias is unlikely to influence our results.

Figures:

Figure 1: The color scale is such that although the shallower areas are orange, everywhere else of interest is red and invisible. Perhaps include a few contour lines or change the color scale so that the defining bathymetry of the troughs can be seen.

Changed colour-scale.

Figure 2: Suggest including the grid in both directions to enhance comparisons of salinities. Axes tick labels are too small.

done

A figure or subpanel showing the quantified trends for this analysis including estimates of uncertainties would substantially improve the grounds for the statements made in the text. I played around with numbers that I pulled from the manuscript figure 2 and got the results shown in the attached figure.

Added the figures in supplementary material.

Figure 3: Is it sensible to average in pressure space? I would like to see a figure or even just statement indicating the flatness of the isopycnals in the regions of interest.

Within the deep layers averaged to construct the time series, density is nearly uniform, so averaging values at constant pressure or constant density would give very similar results.

Figure S1: To recognize the similarities and differences in these time series it would be useful to have these plots either distributed vertically, or better yet, all in the same figure (yes, the latter would be more complicated as it would require multiple axes).

We have changed the figure (now Figure S5) in order to have the plots distributed vertically.

Also, what are the definitions of severity and intensity? It is interesting that they are so different.

The severity is the sum of the differences between the measured wind speed and the relative percentile level. The intensity is the ratio between severity and duration. Thus, the severity score highlights the magnitude of a katabatic event, measured in terms of deviation from the reference percentile level. While, the intensity score provides information about variability and trends of strong and short events.

We have added the sentence: "Thus, the severity score highlights the magnitude of a katabatic event, measured in terms of deviation from the reference percentile level. While, the intensity score provides information about variability and trends of strong and short events to the section S.3 In-situ wind observations.

Also, same as above, are the trends significant given the strong variability?

We have also added the trend significances: The trend for frequency and duration are significant at 99%, while the trends for intensity and severity are not significant.

”

Reviewer #2 (Remarks to the Author):

Dense water export from the Ross Sea is an important issue. The fact that salinity has been increasing since 2014/2015 and is represented by observations from more than one year is important.

There should be some indication somewhere in the paper of the times when data were collected. The missing symbols in Fig 2 tell the story, but it would be nice to have a simple table somewhere indicating which years have observations from each of the regions, maybe give the month of each year.

We have added more information on the dataset and a table indicating the number and date range of the casts that have been used to construct an average in each austral summer, in each region.

When during the year were these observations made? I assume sometime during Nov to Feb of the indicated year. Given the page constraints, maybe a simple statement that most measurements

were in early? mid? late? summer with the explicit times in the supplemental information? This timing during

done

the year will only affect the surface salinity in any case.

I don't agree that surface processes don't affect deep salinity (lines 104-106). Deep convection in winter over most of the Ross Sea takes surface conditions throughout the water column. So current deep water salinity can certainly be affected by surface salinity in the previous summer as well as import of water from the Amundsen shelf (although with a longer delay than one year)

This sentence has been removed. (We agree that surface processes can affect deep properties. Our point was that it was hard to explain synchronous freshening throughout the water column by appealing to surface processes alone. Processes like vertical mixing or deep convection can transfer surface properties downward, but with some delay. As this sentence was not clear and is not central to the argument made, we have removed it).

The conclusions of the paper are a bit nebulous: might be winds, might be ice export, might be Amundsen Sea import, might be CDW import, etc. I think there is enough room in this paper for a few plausibility estimates. How much would sea ice export need to change to produce the indicated salinity increase? How much would meltwater in the Amundsen need to decrease to reduce meltwater import sufficiently to produce the indicated change? Would more CDW import make a difference? Combination of these effects? These estimates might add a paragraph to the paper and at least point readers to likely processes.

We have now included some plausibility estimates. We calculated the increase in salt content of the HSSW layer. We then calculated how much sea ice formation would be needed to add this much salt, or how much freshwater would need to be removed to account for the increase in salinity. We conclude that a substantial but plausible anomaly in either brine rejection or freshwater input could account for the increase in salinity. It is not possible to be more definitive in the absence of measurements of the key terms in the salt budget of the Ross Sea.

I am confused about how the contours in Fig 3 were created. There seem to be extrema in salinity during some years when there are no observations. There is also a bit of structure in the contours between observations. The authors need to explain how this is possible or what additional information is used to create these structures.

The small structure in the contours between observations was created by the interpolation method, but the general pattern is not influenced. In any case we have re-made and improved the figures.

The new figures, do not show extrema in salinity.

Fig 2: There should be different color choices for TNB and JT curves. Although the higher values should clearly be in Terra Nova Bay, so with a bit of thought the reader should be able to tell.

changed

Reviewer #3 (Remarks to the Author):

The authors identify a rapid rebound of salinity in the Ross Sea, after a previously well-documented minimum was reached in the period 2012-2014. Identifying changing salinity in this region is important as the salinity of HSSW helps determine the production rate and water mass characteristics of Ross Sea Bottom Water (RSBW), a major contributor to Antarctic Bottom Water (AABW) that is the largest abyssal water mass in the global ocean.

Assuming that the results are robust (see Major Comments, below), the observational result justifies publication in Nature Comms., with the potential to be motivational to further research and, therefore, be well cited. The paper is short, and could even be shortened further since the paragraph devoted to Conclusions (lines 138-158) basically restates material that has just been reported in the

previous paragraphs. However, the analysis of causality to explain the observation requires much more care: also see Major Comments, below.

MAJOR COMMENTS

1. Much more information is needed on sampling per region per year: Information provided about the CTD data is totally inadequate for assessing whether the results might have arisen from sampling biases. How many CTD profiles are included in each region in each sampled year? Is the spatial distribution of each set consistent with the assumption that mean profile values can be reasonably compared? Given that interannual (and annual) variations are large in Ross Sea hydrographic conditions, might apparent changes in hydrography for a specific region be caused by motion of fronts or water mass property gradients, and/or a different “centroid” of sampling? The authors might argue that the error bars on Figure 1 “answer” these questions, but they don’t (for me) and in any case they not explained.

We have added more information about the dataset in the supplementary information, including the number and date range of the profiles used for computing the austral summer average in each region. In each case, we have only used stations that are closely spaced and from the same season.

We have added a new paragraph in the supplementary material (S2) where we address the concerns about sampling bias. The homogeneity of the data set is reflected in the small standard deviation on pressure surfaces of the profiles that have been averaged for each austral summer, showing that spatial bias is unlikely. We use moored time series from TNB to confirm that temporal variability during austral summer is small and therefore there is unlikely to be any temporal bias. See S2 for a more extensive reply.

At DT mouth, the area most affected by movements of the Antarctic Slope Front, we have now selected only profiles that have neutral density more than 28.27 and temperature below -1.85 (following Orsi & Wiederwoh 2009), to ensure we only sample the outflow of HSSW. This removes an artefact introduced by including some MCDW in the earlier plots and ensures the time series from each region are consistent.

2. The authors conclude that changes in freshwater input from the Amundsen Sea explains the changes, but this is mainly based on (i) observed changes in Amundsen Sea ice sheet mass loss rates from Konrad et al. 2017 and a couple of papers looking at single ice shelves and grounded-ice catchments and (ii) rejecting one specific hypothesis – that variability of some aspect of katabatic winds in Terra Nova Bay (TNB) could drive changes in dense water production – which they reject on the basis of Figure S1 that TNB katabatics are *not* responsible, even though they don’t present a specific mechanistic link between individual katabatic metrics and salinity. I believe the choice of hypothesis to reject is based on the availability of a katabatics record from an AWS, not on being confident that it is the only alternative to the Amundsen Sea advection freshwater source.

Many authors have shown the link between TNB polynya and the katabatic winds (Bromwich and Kurtz, 1984; Kurtz and Bromwich, 1985; Van Woert, M.L., 1999), and in particular Rusciano et al 2013 showed the link between the persistence of the katabatic wind, the opening of the polynya and HSSW. They highlight that the HSSW salinity produced in the

TNB polynya is not only related to the intensity of the katabatic winds but above all to the duration and the number of the katabatic events which reach the bay.

We have added a sentence in the paper: "Has been shown that the activity of the TNB polynya depends on the katabatic winds (Bromwich and Kurtz, 1984; Kurtz and Bromwich, 1985; Van Woert, M.L., 1999b.), Rusciano et al 2013 highlight the link between the persistence of the katabatic wind the opening of the polynya and the HSSW formation"

My intuition says that the authors are correct, but then why put so much effort into one specific negative hypothesis (katabatics in TNB) rather than either also dismissing other hypotheses (Ross Shelf Polynya variability? Freshwater from Ross Ice Shelf? Residual effects of the 2001-2005 iceberg "blockage" around Ross Island, given retention time scales on the continental shelf? ???) or doing the best possible analyses for the "positive" hypothesis of Amundsen Sea freshwater? For example:

a) Show that the change in Ross Sea salinity is consistent with the magnitude of variability in the freshwater equivalent of the Amundsen Sea ice sheet mass balance (and/or sea ice advected westward);

b) Look at sea ice motion vectors as a guide to changes in annual flux of Amundsen Sea waters into the eastern, and subsequently all of the Ross Sea.

c) Look at the weather patterns (from reanalyses like ERA-Int etc.) to seek evidence that ocean flows from the Amundsen to the Ross should have decreased in the last few years.

d) Create (and show) sea ice metrics that make sense in view of proposed mechanisms. You state that sea ice is acting the wrong way to explain the recent trend in salinity, but perhaps there is an EOF or other analysis that pops out the signal you want to see. There are recent papers, some of which you cite (e.g., cites 28-31) that discuss some of the "external" mechanisms responsible for recent rebounds in Antarctic climate. These should probably be re-read and used to provide a more robust defense of the mechanistic relationship proposed between the Amundsen Sea and the Ross salinity.

In the revision, we have followed the advice of Reviewer 2 and now provide estimates of how much sea ice formation and/or freshwater input would need to change to explain the "rebound" in salinity. We first calculate the change in salt content of the HSSW layer (using the volume of HSSW according to Orsi and Wiederwohl (2009)). We then calculate how much sea ice would need to form to deliver this much salt. We also assess how large a decrease in freshwater input reaching the HSSW layer would be needed to explain the change in salinity. We find that the magnitude of the anomalies required are large but in the ballpark. (Given that the salinity anomaly is large relative to the variability in the pre-2014 record, it is not a surprise that large anomalies in salt or freshwater input are needed to explain them.) We conclude that an increase in sea ice formation and/or a reduction in freshwater input can likely account for the change in HSSW salinity.

This "conclusion" is admittedly weak. But a more definitive statement requires quantitative estimates of the terms contributing to the salt budget of the Ross Sea, and their variation in time. These estimates do not exist. (We note that the widely-accepted attribution of the pre-2014 freshening trend to increased melt input from the Amundsen Sea is based on a similar consistency argument, rather than a quantitative analysis of the salt budget.)

The calculations suggested by the reviewer would still fall short of a conclusive quantitative demonstration of the mechanism responsible for the salinity increase. A time series of sea ice drift vectors would provide information on sea ice motion, but cannot be translated into freshwater flux without knowledge of sea ice thickness. Reanalyses would provide information on wind changes, but it is not clear to us how to use this to calculate velocities and freshwater transport in the absence of salinity measurements. A more complete analysis of changes in sea ice formation and its relationship to changes in winds and other factors is possible (and underway), but is beyond the scope of the present paper.

3. Lines 103-106. You comment that “local processes affecting the surface layer” cannot explain freshening below the surface layer. This is not true. If diapycnal mixing is important than the fresh signal propagates downward. Dumping more fresh water into the SML in summer also reduces the salinity to which subsequent winter convection might drive full-depth averaged salinity (e.g., in the central Ross Sea, there is interannual variability in the depth of the winter mixed layer.) Furthermore, there are feedbacks between circulation and stratification so that, while a fresher cap might prevent the deeper water from mixing with the SML in a 1D sense, there might be a change in the 3D flow field that causes correlated changes in deep hydrography.

This sentence has been removed.

MINOR COMMENTS

Lines 38, 39, maybe elsewhere: You can't talk of “freshened by” then give a negative number; same for “decrease in salinity”.

changed

Figure 2: Colors for TNB and JT are too close together. I could only sort them out by knowing that TNB salinity should be higher.

changed

Lines 114-115: => “are believed to make smaller contributions to”.

changed

Lines 138-158: Given how short the paper is, even though it would be longer if you follow by Major Comments, a simple restatement of what has already been said doesn't seem worthwhile. Maybe expand upon the “consequences” paragraph, which is now a short final paragraph here.

Given the presence of a large ice shelf, be pedantic and always say either "CONTINENTAL shelf" or "ICE shelf" when you refer to "shelf".

Done

-- Laurie Padman

Reviewers' comments:

Reviewer #1 (Remarks to the Author):

Thank you for the care you have taken in answering each of my comments not only with words but also with analysis and figures to back the words up. I carefully read through your responses and then reread the manuscript from beginning to end. It read very well. I believe that it is much improved and ready for publication. I look forward to seeing what future observations tell us about is happening in this local region whose waters have such a broad, indeed global scope of influence.

A couple of pieces of trivia:

The sentence beginning on line 142 and end on Line 146: the words "oxygen isotope data" appear twice. It looks like the sentence was being rearranged but did not quite get tidied up.

The term "crude" is used twice to describe the some of calculations. I would suggest rather using the word "rough" or "back-of-the-envelope" as they are not quite so derogatory.

Reviewer #2 (Remarks to the Author):

This is an important and compelling paper. The modifications by the authors clarify the various questions that I was concerned about. They also seemed to address the questions raised by the other reviewers.

In addition, the new information in the supplementary details make the analysis process clear and contribute to the compelling nature of the paper.

I feel that the paper is suitable for publication in its present form (both the paper and supplementary information).

I agree to reveal my identity to the authors:

John Klinck

Reviewer #3 (Remarks to the Author):

Review 2 of Castagno et al., "Rebound of shelf water salinity in the Ross Sea", submitted to Nature Comms.

The authors identify a rapid rebound of salinity in the Ross Sea, after a previously well-documented minimum was reached in the period 2012-2014. Identifying changing salinity in this region is important as the salinity of HSSW helps determine the production rate and water mass characteristics of Ross Sea Bottom Water (RSBW), a major component of Antarctic Bottom Water (AABW) that is the largest abyssal water mass in the global ocean.

This is my second review of this paper. I thank the authors for trying to take into account all the concerns of the three original reviewers. The new version is much more careful about what the results do and don't show. As I said last time, the paper is sufficiently novel to satisfy requirements for Nature Comms., with the potential to be motivational to further research and, therefore, well cited.

However, I still have several problems with the new version. These are mostly "minor", in that I trust the science more now, and my main problems are with presentation. Nevertheless, I don't think the paper is ready to be accepted.

-- Laurie Padman

MAJOR COMMENTS

1. You need, probably as the first Supp figure, a T-S plot with water masses identified on it. This would clarify the relationship between HSSW, RSBW and AABW, and make it easier to talk about CDW and other aspects of the continental-shelf budgets.

20-21: The statement “The rate, magnitude ... in the observational record.” is correct. However, the pre-1990s record is sparsely sampled, so these rates could have easily happened before, between cruises.

Figures 2 and 3 give me a different view of what you should be explaining. Prior to 2008, there is probably no significant trend in any area, with the possible exception of TNB. So, rather than a “rebound”, I see a need to explain the freshening, predominantly in 2012-2014, which looks more like an anomaly than the rebound does.

The new salt budget calculations are really good. It doesn't worry me that we don't know which is causing the salinity changes; these calculations will be very motivational to modelers. However, I don't think it is well organized. I recommend three things: (a) Put all the salt budget material in the Discussion. Start with a general comment like “The salt content of the HSSW layer can change from the following processes: ..., ..., and” Then work through everything in approximate order of likely importance, e.g., Amundsen Sea, sea ice, Ross Ice Shelf, TNB katabatics, ??? (b) Add a table, comparing the possible source magnitudes. Then Discussion would end with what future measurements would be important to improve our understanding of HSSW salinity changes. Note that there are time series of sea ice volume variability, from ice velocity and assumed ice thickness (Comiso, Paul Holland and others), and ICESat altimetry (Kurtz and Markus) and maybe new products based on CryoSat-2. The melting variability of Amundsen Sea ice shelves is known, e.g., the Paolo et al. 2018 paper that you already cite.

The point, with most prior analyses of all these existing data sets, is that there are known ties between sea ice (Stammerjohn et al., Raphael et al., P. Holland et al.) and climate, and ice shelves (Paolo et al., 2018, Dutrieux et al. 2014) and climate, that indicate the factors that can contribute to salt fluxes. For example, perhaps you can hypothesize that the Southern Annular Mode, ENSO, or Interdecadal Pacific Oscillation are important. There are known, documented ties between climate indices and atmospheric forcing, e.g., a recent paper relating warm air fluxes south into the eastern Ross Sea from a combination of El Nino and SAM forcing (really, magnitude and position of the Amundsen Sea Low). If you plotted these time series, plus, say, sea ice, you might find that the “rebound” starting around 2015 is really correlated with the ENSO/SAM effects that show up in Amundsen Sea ice shelves basal melt, sea ice and atmosphere. An alternative view might be that the 2012-2014 salinity “dip” is La Nina. 2000 is also a strong La Nina, and it is another place where HSSW salinity declines.

I would not expect the correlations to be really clear. But I do think your paper should show the relationship (even if not clear) between the processes and your results. This puts some physics in a mainly statistics paper, and makes the Discussion about possible salt sources more structured.

There is a new Ross Sea paper that you might find interesting:

<https://agupubs.onlinelibrary.wiley.com/doi/full/10.1029/2018JC014683>. It does not focus on HSSW, or even on the areas that you are interested in in the western Ross, but it does discuss freshwater input from the Amundsen, and the Supplementary Information figures S3 and S4 show 3-year full-depth T and S plots.

There is a new paper on time dependence of Pine Island Ice Shelf basal melting: <https://www.the-cryosphere-discuss.net/tc-2018-209/>. Not sure that this is useful itself, but the citations in it might be worth looking at.

The mass balance of *grounded* ice sheets as a function of time, up to the end of 2017, is in Shepherd et al. 2018 (<https://doi.org/10.1038/s41586-018-0179-y>). Figure 1b might be interesting for you.

60-66: Rather than citing the significance level, I would prefer to see a "+/-" error on the slope, where you explain that "Cited errors are the 95% confidence intervals."

I always have trouble with averages of specific water masses. You have explained the procedure, so it's okay here. But if a water mass "cools", sometimes it is because its distribution in T-S space has broadened and some previously included water is being left out (or vice versa) by the thresholding, not because the total heat has changed.

89: "large and rapid in comparison with salinity trends": This is only true if you only focus on *increasing* trends. In fact, the magnitude is comparable to the *decline* leading into the 2012-2014 minimum. This is a major reason why I see the 2012-2014 minimum to be the primary anomaly worth discussing, not the post-2014 "rebound".

90-94: I don't like comparing trends from a short record (4 years post-2014) with those from a long record (1995-2014) in this way. (a) Given that there is a lot of interannual, this is not surprising. (b) a better comparison would be with *magnitudes* of other 4-year trends. Then you'd see that it is similar (in magnitude; opposite in sign) to 2008-2012.

I assume the plan is to get all 4 panels of Fig. 3 onto 1 page. Then, there should be a colorbar, and a colorscale that is constant for all panels.

136: I think the “Discussion” section is where you should put all the attempts to explain the signal; see earlier Major Comments.

145-148: This conclusion will be much stronger once all the discussion of potential salt sources are in the same place, preferably with a Table. See, e.g., Table 2 in Porter et al. 2019 as a possible way to summarize uncertain source contributions to observations.

179-181: Again you are mixing time scales. The “recent decades” trend in meltwater from WAIS is consistent with the freshening trend in HSSW on the same time scale of “recent decades”. For this comparison to be useful, you would need to compare WAIS freshwater trends on the post-2014 time scale.

190-195: This is a very complex sentence. Two things: (a) It usually is cleaner if you do **not** start a sentence with “While ...”, especially if it is long. (b) Again ... the “rebound” is really, primarily, just the recovery from the large 2012-2014 anomaly. My problem with discussing the rebound instead of the anomaly is that it means you focus on comparing long time scales (with lots of short-timescale interannual signal) with just the short time scale representative of the interannual variations we could already see from the pre-2014 period.

196-198: This sentence is confusing because it makes it sound like RSBW is something different from just a component of AABW. Also, HSSW is not really a precursor to AABW: it is a component of it, the other being CDW. It would be better here if you laid it out more clearly, that saltier HSSW mixing with CDW will produce denser RSBW, a major component of global AABW, assuming that CDW and mixing processes along the Ross Sea continental slope remain the same.

MINOR COMMENTS

11. (a) I think RSBW is a **component** of AABW, not a “source”. (b) I think you need to tell us what the largest component is (Weddell Sea).

Check whether acronyms are allowed in the bold “Abstract” paragraph. Regardless, you need to make sure every acronym is explained when it first comes up. E.g., CDW on line 26 is not explained while, on line 32, HSSW *has* already been explained.

27-28: You claim that “In recent decades AABW has warmed⁴, freshened^{1,4}, and decreased in volume and density⁵, contributing to the increase in global ocean heat content and sea level rise^{1,2}”. This isn’t totally obvious. How does a *decrease* in volume lead to an *increase* in sea level? Presumably the *net* effect is for an increase, through warming, but needs to be explained.

85-86: lower-case “k” in “kg” (for kilograms)

134-135: I do *not* like the structure “... with fresh (salty) fresh (salty) ...” where I am meant to understand that the words in brackets are the opposite sense. If you say, instead “with fresher ... with fresher ... and ... deeper ...” then the opposite case can be assumed to be true anyway.

139-143: I don’t think you need to say “oxygen isotope data” twice. If you do, I am not understanding this sentence.

143-145: This idea is supported nicely by Porter et al. 2019, JGR-Oceans; URL provided in Major Comments.

We thank the reviewers for their comments, which have helped to further improve the paper. In our response, the reviewers' comments are in italics, our response in red.

Reviewer #1 (Remarks to the Author):

Thank you for the care you have taken in answering each of my comments not only with words but also with analysis and figures to back the words up. I carefully read through your responses and then reread the manuscript from beginning to end. It read very well. I believe that it is much improved and ready for publication. I look forward to seeing what future observations tell us about is happening in this local region whose waters have such a broad, indeed global scope of influence.

Thank you for your contribution to improving the manuscript.

A couple of pieces of trivia:

The sentence beginning on line 142 and end on Line 146: the words "oxygen isotope data" appear twice. It looks like the sentence was being rearranged but did not quite get tidied up.

Corrected.

The term "crude" is used twice to describe the some of calculations. I would suggest rather using the word "rough" or "back-of-the-envelope" as they are not quite so derogatory.

We have substituted "rough" for "crude," as suggested.

Reviewer #2 (Remarks to the Author):

This is an important and compelling paper. The modifications by the authors clarify the various questions that I was concerned about. They also seemed to address the questions raised by the other reviewers.

In addition, the new information in the supplementary details make the analysis process clear and contribute to the compelling nature of the paper.

I feel that the paper is suitable for publication in its present form (both the paper and supplementary information).

I agree to reveal my identity to the authors:

John Klinck

We thank Professor Klinck for his thoughtful comments that helped us improve the paper.

Reviewer #3 (Remarks to the Author):

Review 2 of Castagno et al., "Rebound of shelf water salinity in the Ross Sea", submitted to Nature Comms.

The authors identify a rapid rebound of salinity in the Ross Sea, after a previously well-documented minimum was reached in the period 2012-2014. Identifying changing salinity in this region is important as the salinity of HSSW helps determine the production rate and water mass characteristics of Ross Sea Bottom Water (RSBW), a major component of Antarctic Bottom Water (AABW) that is the largest abyssal water mass in the global ocean.

This is my second review of this paper. I thank the authors for trying to take into account all the concerns of the three original reviewers. The new version is much more careful about what the results do and don't show. As I said last time, the paper is sufficiently novel to satisfy requirements for Nature Comms., with the potential to be motivational to further research and, therefore, well cited.

However, I still have several problems with the new version. These are mostly "minor", in that I trust the science more now, and my main problems are with presentation. Nevertheless, I don't think the paper is ready to be accepted.

-- Laurie Padman

We appreciate Dr Padman's thorough and careful assessments of the manuscript, which have helped to further improve the clarity of the text.

MAJOR COMMENTS

1. You need, probably as the first Supp figure, a T-S plot with water masses identified on it. This would clarify the relationship between HSSW, RSBW and AABW, and make it easier to talk about CDW and other aspects of the continental-shelf budgets.

The data we present in the paper is limited to the continental shelf, so a T-S plot of our data will not include RSBW and AABW (which are not found on the shelf). The paper cited in the sentence that describes how HSSW and CDW contribute to formation of AABW (citation 10) includes the T-S plot suggested. We believe this is sufficient and avoids any confusion that might be caused by introducing a new data set here that is only used in a T-S plot provided for context.

20-21: The statement "The rate, magnitude ... in the observational record." is correct. However, the pre-1990s record is sparsely sampled, so these rates could have easily happened before, between cruises.

We have altered the final sentence of the abstract as follows: “The rate, magnitude and duration of the recent salinity increase are unusual in the context of the (sparse) observational record”.

Figures 2 and 3 give me a different view of what you should be explaining. Prior to 2008, there is probably no significant trend in any area, with the possible exception of TNB. So, rather than a “rebound”, I see a need to explain the freshening, predominantly in 2012-2014, which looks more like an anomaly than the rebound does.

We are a little puzzled by this comment. This description of the time series (little change, then a sudden freshening in 2012-2014, then recovery to earlier values) is not consistent with the data. At TNB, the trends prior to 2014 are not sensitive to the end points chosen, as seen in the table below (values are change in salinity per decade). In addition, the trends prior to 2014 are similar to those found by Jacobs and Giulivi (2010). Including the years 2014-2018 reduces the magnitude of the trend by about half, underscoring the anomalous nature of the rebound in salinity after 2014.

	1995-2008	1995-2012	1995-2014	1995-2018
TNB	-0.039 ($R^2 = 0.44$)	-0.042 ($R^2 = 0.62$)	-0.045 ($R^2 = 0.74$)	-0.023 ($R^2 = 0.30$)

We have modified the text to make it clear that the time series of most relevance to a discussion of HSSW properties is TNB, where the saltiest HSSW is found and from where dense water is exported through the DT to reach the deep ocean. (“The most relevant time series for HSSW is TNB, where the saltiest and densest HSSW is found. DT, through which the dense HSSW is exported to the continental slope, is also highly relevant, but closer to the shelf break and therefore more influenced by factors unrelated to HSSW formation.”)

The TNB time series is also the most complete and extends to 2018. The trends at DT and JT are more sensitive to the choice of end points but still show a significant freshening trend. (e.g. at DT, the 1995-2008 trend is -0.02 dec^{-1} , compared to -0.043 dec^{-1} for the 1995-2014 period).

The data are not consistent with the assertion that there is probably no significant trend in any area, with the possible exception of TNB. So, rather than a “rebound”, I see a need to explain the freshening, predominantly in 2012-2014, which looks more like an anomaly than the rebound does.

The new salt budget calculations are really good. It doesn't worry me that we don't know which is causing the salinity changes; these calculations will be very motivational to modelers. However, I don't think it is well organized. I recommend three things: (a) Put all the salt budget material in the Discussion. Start with a general comment like “The salt content of the HSSW layer can change from the following processes: ..., ..., and” Then work through everything in approximate order of likely importance, e.g., Amundsen Sea, sea ice, Ross Ice Shelf, TNB katabatics, ??? (b) Add a table, comparing the possible source magnitudes. Then Discussion would end with what future measurements would be important to improve our understanding of HSSW salinity changes. Note that there are time series of sea ice volume variability, from ice velocity and assumed ice thickness (Comiso, Paul Holland and others), and ICESat altimetry (Kurtz and Markus) and maybe new products based on CryoSat-2. The melting variability of Amundsen Sea ice shelves is known, e.g., the Paolo et al. 2018 paper that you already cite.

We have revised the Discussion section. We have also added a citation of Porter et al. (2019). While estimates of sea ice drift exist, ice thickness is unknown (the Kurtz and Markus record is only 5 years long) and highly variable, hence estimates of sea ice volume change are subject to large uncertainty. There are estimates of variability in melt of Amundsen Sea ice shelves, but the fraction that reaches the Ross Sea HSSW and the time it takes to get there are poorly known. Given these large uncertainties, we believe the back-of-the-envelope calculations we have included are more useful than a table of estimates with large but unknown uncertainties.

The point, with most prior analyses of all these existing data sets, is that there are known ties between sea ice (Stammerjohn et al., Raphael et al., P. Holland et al.) and climate, and ice shelves (Paolo et al., 2018, Dutrieux et al. 2014) and climate, that indicate the factors that can contribute to salt fluxes. For example, perhaps you can hypothesize that the Southern Annular Mode, ENSO, or Interdecadal Pacific Oscillation are important. There are known, documented ties between climate indices and atmospheric forcing, e.g., a recent paper relating warm air fluxes south into the eastern Ross Sea from a combination of El Nino and SAM forcing (really, magnitude and position of the Amundsen Sea Low). If you plotted these time series, plus, say, sea ice, you might find that the “rebound” starting around 2015 is really correlated with the ENSO/SAM effects that show up in Amundsen Sea ice shelves basal melt, sea ice and atmosphere. An alternative view might be that the 2012-2014 salinity “dip” is La Nina. 2000 is also a strong La Nina, and it is another place where HSSW salinity declines.

I would not expect the correlations to be really clear. But I do think your paper should show the relationship (even if not clear) between the processes and your results. This puts some physics in a mainly statistics paper, and makes the Discussion about possible salt sources more structured.

We agree that gaps remain in our understanding of the physical mechanisms that drive variability of Ross Sea shelf waters. We furthermore agree that ENSO, SAM, IPO, and the ASL all are likely to influence the salinity of shelf waters in the Ross Sea. But this is a large and complex topic, with a literature that is not entirely consistent (e.g. the 2016 decline in Antarctic sea ice has been attributed to many different drivers). While there are certainly “known, documented ties between climate indices and atmospheric forcing,” the links between climate indices and HSSW salinity are less clear and difficult to unravel in a definitive way in the absence of quantitative knowledge of the contributions to the salt budget. The literature on this topic suggests the Ross Sea shelf waters are unlikely to respond to a single climate mode, and so a plot of HSSW salinity versus the climate indices will not be very informative.

As noted by Reviewers 1 and 2, the strength of the paper is that it provides a concise, quantitative analysis of a unique time series of global relevance that highlights recent changes that are unusual in the context of the long-term observational record - changes that are likely to have broader implications, given the importance of this region for ventilation of the abyssal ocean. We are reluctant to dilute the message of the paper by adding additional discussion of drivers that cannot be assessed with the information we have in hand, and thus will inevitably be inconclusive.

There is a new Ross Sea paper that you might find interesting:

<https://aqupubs.onlinelibrary.wiley.com/doi/full/10.1029/2018JC014683>. It does not focus on HSSW, or even on the areas that you are interested in in the western Ross, but it does discuss

freshwater input from the Amundsen, and the Supplementary Information figures S3 and S4 show 3-year full-depth T and S plots.

Thanks for the citation. We have now cited Porter et al. in several places in the manuscript.

There is a new paper on time dependence of Pine Island Ice Shelf basal melting: <https://www.the-cryosphere-discuss.net/tc-2018-209/>. Not sure that this is useful itself, but the citations in it might be worth looking at.

The mass balance of *grounded* ice sheets as a function of time, up to the end of 2017, is in Shepherd et al. 2018 (<https://doi.org/10.1038/s41586-018-0179-y>). Figure 1b might be interesting for you.

Thanks for the additional references (we now cite Shepherd et al. 2018, see response below to line 179-181).

In our view, the difficulty of directly relating changes in melt and ice discharge to time-variability of HSSW salinity is lack of knowledge of the fate of melt water added in the Amundsen Sea: how much of the melt makes it to the Ross Sea (as opposed to mixing with offshore waters, or moving west in the Antarctic Slope Front rather than on the shelf), when does it arrive, and what fraction reaches the HSSW layer? Model studies like that of Nakayama provide a clue but fall short of the information needed to relate variability of HSSW salinity to variability of ice shelf melt/discharge at multiple glaciers. What is needed is the integral over all glaciers of the convolution of the melt time series for each individual glacier and a function that represents the transport from that particular glacier to the HSSW layer in the western Ross Sea. This information does not exist. (A quantitative salt budget would also require knowledge of freshwater input from icebergs and sea ice, and their melt – also unknown).

60-66: *Rather than citing the significance level, I would prefer to see a “+/-” error on the slope, where you explain that “Cited errors are the 95% confidence intervals.”*

The significance level of a trend and the uncertainty in the slope are both useful, so we have now included both pieces of information. Stating the level of significance associated with a trend is consistent with the approach used in prior studies of change in properties of Ross Sea shelf water (e.g. Jacobs and Giulivi, 2010).

We have also changed Figure S4. Now we have plotted the 95% confidence interval instead of the 95% prediction interval.

I always have trouble with averages of specific water masses. You have explained the procedure, so it's okay here. But if a water mass “cools”, sometimes it is because its distribution in T-S space has broadened and some previously included water is being left out (or vice versa) by the thresholding, not because the total heat has changed.

As noted, this is not an issue here (we average over a depth interval, not a water mass classification defined by an arbitrary threshold).

89: *“large and rapid in comparison with salinity trends”*: This is only true if you only focus on **increasing** trends. In fact, the magnitude is comparable to the **decline** leading into the 2012-2014 minimum. This is a major reason why I see the 2012-2014 minimum to be the primary anomaly worth discussing, not the post-2014 “rebound”.

The statement that the rate of decline leading into the 2012-2014 minimum is as large as the increase post-2014 is not supported by the data. At TNB, the decrease in salinity from 2008 to 2012 is 0.0467 in 4 years (0.0117 per year). The salinity increase from 2014 to 2018 is 0.0860 in 4 years (0.0215 per year), almost twice as rapid. At DT, the salinity decline from 2008 to 2012 is 0.0587 in 4 years (0.0147 per year), while the increase from 2014 to 2017 is 0.0778 in 3 years (0.0259 per year).

90-94: *I don’t like comparing trends from a short record (4 years post-2014) with those from a long record (1995-2014) in this way. (a) Given that there is a lot of interannual, this is not surprising. (b) a better comparison would be with *magnitudes* of other 4-year trends. Then you’d see that it is similar (in magnitude; opposite in sign) to 2008-2012.*

The purpose of this sentence is to answer the question: are the post-2014 values of salinity unusual in the context of the preceding record, which consists of a multi-decadal freshening trend plus interannual variability? We find that the recent observations are not consistent with the pre-2014 record (i.e. they lie more than 5 standard deviations above the long-term trend, where the standard deviation is calculated from the de-trended pre-2014 record and is therefore a measure of interannual variability). As noted above, the rate of decrease in salinity from 2008 to 2012 is about half the rate of salinity increase after 2014.

To address the concern raised, we have cut the last part of the sentence that compares the rate of change in the past 4 years to the rate of change between 1995 and 2014.

I assume the plan is to get all 4 panels of Fig. 3 onto 1 page. Then, there should be a colorbar, and a colorscale that is constant for all panels.

In this case, it is more useful to have different color scales in each panel because the variability from year-to-year at each site (the primary signal of interest in this plot) is smaller than the difference in salinity between sites. With a common color bar, all that can be inferred from the plot is that TNB is saltier than the other sites, something that is already clear from Figure 2.

136: *I think the “Discussion” section is where you should put all the attempts to explain the signal; see earlier Major Comments.*

The Discussion has been re-organised.

145-148: *This conclusion will be much stronger once all the discussion of potential salt sources are in*

the same place, preferably with a Table. See, e.g., Table 2 in Porter et al. 2019 as a possible way to summarize uncertain source contributions to observations.

This statement is a summary of the findings reached in the papers cited (now including Porter et al.), not a conclusion of the present work.

179-181: Again you are mixing time scales. The “recent decades” trend in meltwater from WAIS is consistent with the freshening trend in HSSW on the same time scale of “recent decades”. For this comparison to be useful, you would need to compare WAIS freshwater trends on the post-2014 time scale.

We agree this sentence needs revision. We now use the 5-year average mass loss estimates from Shepherd et al. (2018) to assess recent changes in mass loss from the WAIS. The sentence now reads “However, mass loss from the West Antarctic Ice Sheet has increased in each of the past three pentads (from $-65 \pm 27 \text{ Gt yr}^{-1}$ in 2002-2007, to $-148 \pm 27 \text{ Gt yr}^{-1}$ in 2007-2012, and $-159 \pm 26 \text{ Gt yr}^{-1}$ in 2012-2017) (Shepherd et al. 2018), which would act to decrease rather than increase the salinity of shelf waters in the Ross Sea.”

*190-195: This is a very complex sentence. Two things: (a) It usually is cleaner if you do *not* start a sentence with “While ...”, especially if it is long. (b) Again ... the “rebound” is really, primarily, just the recovery from the large 2012-2014 anomaly. My problem with discussing the rebound instead of the anomaly is that it means you focus on comparing long time scales (with lots of short-timescale interannual signal) with just the short time scale representative of the interannual variations we could already see from the pre-2014 period.*

This paragraph has been revised to simplify the text:

“The anomalies in sea ice formation or meltwater input needed to account for the observed increase in salinity of HSSW between 2014 and 2018 are large, relative to their mean values and variability, but of the right order of magnitude. These rough calculations therefore suggest that an increase in sea ice formation and/or a reduction in freshwater input could explain the recent increase in HSSW salinity. The salinity increase is unusual in the observational record and requires a climate anomaly of sufficient magnitude to reverse 20 years of freshening at the multi-decadal trend observed prior to 2014. Further work is needed to identify the physical mechanisms responsible for the salinity increase and their link to larger-scale climate phenomena. Contributions to the salt budget of the Ross Sea must be better observed and understood, in particular formation and export of sea ice and inflow of freshwater from the Amundsen Sea.”

196-198: This sentence is confusing because it makes it sound like RSBW is something different from just a component of AABW. Also, HSSW is not really a precursor to AABW: it is a component of it, the other being CDW. It would be better here if you laid it out more clearly, that saltier HSSW mixing with CDW will produce denser RSBW, a major component of global AABW, assuming that CDW and mixing processes along the Ross Sea continental slope remain the same.

While we believe our use of precursor is consistent with the definition of the word, we have altered the text to try and address the concerns raised: “RSBW is formed from a mixture of HSSW and CDW. Just as the multi-decadal freshening of HSSW caused a reduction in the salinity and density of RSBW⁴⁻⁷, the recent shift to saltier HSSW will result in a rebound in the salinity and density of RSBW,

if changes in CDW or mixing do not compensate for the increase in HSSW salinity. If the shift to saltier HSSW and RSBW persists, this will have repercussions for abyssal stratification and ventilation, oceanic CO₂ sequestration, ocean heat storage, and the lower limb of the global overturning circulation.”

Note that in the Introduction, we explain the contribution of HSSW to AABW: “HSSW exported from the continental shelf mixes with CDW as it descends the continental slope, producing AABW.”

MINOR COMMENTS

11. (a) I think RSBW is a **component** of AABW, not a “source”. (b) I think you need to tell us what the largest component is (Weddell Sea).

Our use of “source” is consistent both with the dictionary definition (‘a process by which a particular component enters a system, as plants are a source of atmospheric oxygen’) and with its usage in the oceanographic literature (e.g. Orsi et al., 1999). The phrase “component of AABW,” on the other hand, is often used in the literature to refer to one of the end member water masses that mix to form AABW. As the paper is not about the Weddell Sea, referring to it in the Abstract will confuse more readers than it will enlighten. For these reasons, we prefer the original wording.

Check whether acronyms are allowed in the bold “Abstract” paragraph. Regardless, you need to make sure every acronym is explained when it first comes up. E.g., CDW on line 26 is not explained while, on line 32, HSSW **has** already been explained.

done

27-28: You claim that “In recent decades AABW has warmed⁴, freshened^{1,4}, and decreased in volume and density⁵, contributing to the increase in global ocean heat content and sea level rise^{1,2}”. This isn’t totally obvious. How does a **decrease** in volume lead to an **increase** in sea level? Presumably the **net** effect is for an increase, through warming, but needs to be explained.

Correct, warming causes thermal expansion. The decrease in AABW volume is compensated by an increase in the volume of warmer layers (i.e. downward displacement of isotherms), resulting in warming of the water column and an increase in steric height. This is explained in detail in the references cited. We feel that the concept of warming leading to a rise in sea level is well known and that further explanation here is both unnecessary and interrupts the flow of the Introduction, so we prefer to keep the text as is.

85-86: lower-case “k” in “kg” (for kilograms)

Edited as suggested.

134-135: I do **not** like the structure “... with fresh (salty) fresh (salty) ...” where I am meant to understand that the words in brackets are the opposite sense. If you say, instead “with fresher ... with fresher ... and ... deeper ...” then the opposite case can be assumed to be true anyway.

Edited as suggested.

139-143: *I don't think you need to say "oxygen isotope data" twice. If you do, I am not understanding this sentence.*

Thank you, the second mention has been removed.

143-145: *This idea is supported nicely by Porter et al. 2019, JGR-Oceans; URL provided in Major Comments.*

We agree and have altered the sentence to include a mention of the Porter et al. results ("Float observations of freshening of the summer mixed layer (Porter et al., 2019) and model simulations of meltwater spread also support the inference that freshwater from the Amundsen Sea influences the properties on the Ross Sea continental shelf.")

REVIEWERS' COMMENTS:

Reviewer #3 (Remarks to the Author):

Review of Castagno et al., "Rebound of shelf water salinity in the Ross Sea", submitted to Nature Comms.

Again, I thank the authors for responding to my detailed comments on previous versions. We are not going to agree on everything: in particular, I still think that the signal can still be interpreted (with the exception of TNB) as potentially "steady-state" (with interannual variability) to 2008, followed by a dip and a rebound. My reason for saying this is not to say you're wrong, but to suggest that, when you go looking for climate drivers, you also look for climate signals that would generate a salinity dip during 2008-2014. Focusing just on the post-2014 "rebound" is limiting what you might look for in climate (atmosphere, sea ice, Amundsen Sea ice shelf melting) records.

However, the text is now very clear about what has been found, and most of my nit-picking has been dealt with, so I'm happy to agree with the other reviewers that this paper is almost ready for publication. And, as I said earlier, I think it's a valuable paper and should be well cited.

Minor comments only, not necessarily requiring edits:

1) In your last rebuttal, you said

"The data are not consistent with the assertion that there is probably no significant trend in any area, with the possible exception of TNB. So, rather than a "rebound", I see a need to explain the freshening, predominantly in 2012-2014, which looks more like an anomaly than the rebound does."

Just to be clear: this is **not** what I said. My comment included "Prior to 2008, ..." and "with the possible exception of TNB". These are very important caveats. However, your new, clearer emphasis on the TNB record takes care of this.

2) In your last rebuttal, you said

“The statement that the rate of decline leading into the 2012-2014 minimum is as large as the increase post-2014 is not supported by the data. At TNB, the decrease in salinity from 2008 to 2012 is 0.0467 in 4 years (0.0117 per year). The salinity increase from 2014 to 2018 is 0.0860 in 4 years (0.0215 per year), almost twice as rapid. At DT, the salinity decline from 2008 to 2012 is 0.0587 in 4 years (0.0147 per year), while the increase from 2014 to 2017 is 0.0778 in 3 years (0.0259 per year).”

I was not totally clear in my original comment. My point was – again, with the exception of TNB – that the rebound just brings salinity back to the 2007 level. The fact that salinity recovers faster than the preceding freshening is, itself, interesting, and probably related to the relationship between timescales for advection and for HSSW production from sea ice formation.

My other point, expressed elsewhere, was that the *absolute values of rates* of freshening (2007-2014) and rebound (post-2014) are “comparable” in that they are both much larger than the trend of the full record (1995-2018). So again, singling out the rebound “feature” in the overall trend, while ignoring the “rapid freshening” during 2007-2014, seems biased. Yes, the rebound *rate* is twice the 2008-2012 freshening rate, but the absolute values of both are an order of magnitude larger than the 23-year trend.

-- Laurie Padman

We thank Laurie Padman for his comments, which have helped to further improve the paper. In our response, the reviewer' comments are in italics, our response in red.

Reviewer #3 (Remarks to the Author):

Review of Castagno et al., "Rebound of shelf water salinity in the Ross Sea", submitted to Nature Comms.

Again, I thank the authors for responding to my detailed comments on previous versions. We are not going to agree on everything: in particular, I still think that the signal can still be interpreted (with the exception of TNB) as potentially "steady-state" (with interannual variability) to 2008, followed by a dip and a rebound. My reason for saying this is not to say you're wrong, but to suggest that, when you go looking for climate drivers, you also look for climate signals that would generate a salinity dip during 2008-2014. Focusing just on the post-2014 "rebound" is limiting what you might look for in climate (atmosphere, sea ice, Amundsen Sea ice shelf melting) records.

We agree that the interannual variations (including, but not limited to, the "dip" after 2008) are interesting signals that potentially provide clues to the physical mechanisms driving variability and trends in Ross Sea shelf water.

However, the text is now very clear about what has been found, and most of my nit-picking has been dealt with, so I'm happy to agree with the other reviewers that this paper is almost ready for publication. And, as I said earlier, I think it's a valuable paper and should be well cited.

Minor comments only, not necessarily requiring edits:

1) *In your last rebuttal, you said*

"The data are not consistent with the assertion that there is probably no significant trend in any area, with the possible exception of TNB. So, rather than a "rebound", I see a need to explain the freshening, predominantly in 2012-2014, which looks more like an anomaly than the rebound does."

*Just to be clear: this is *not* what I said. My comment included "Prior to 2008, ..." and "with the possible exception of TNB". These are very important caveats. However, your new, clearer emphasis on the TNB record takes care of this.*

The text we quoted included the second important caveat (see highlight above). We believe the preceding three paragraphs of our response make it clear that we understood the "prior to 2008" caveat and took it into account in our response(e.g. calculating trends prior to and post 2008).

2) *In your last rebuttal, you said*

"The statement that the rate of decline leading into the 2012-2014 minimum is as large as the increase post-2014 is not supported by the data. At TNB, the decrease in salinity from 2008 to 2012 is 0.0467 in 4 years (0.0117 per year). The salinity increase from 2014 to 2018 is 0.0860 in 4 years

(0.0215 per year), almost twice as rapid. At DT, the salinity decline from 2008 to 2012 is 0.0587 in 4 years (0.0147 per year), while the increase from 2014 to 2017 is 0.0778 in 3 years (0.0259 per year)."

I was not totally clear in my original comment. My point was – again, with the exception of TNB – that the rebound just brings salinity back to the 2007 level. The fact that salinity recovers faster than the preceding freshening is, itself, interesting, and probably related to the relationship between timescales for advection and for HSSW production from sea ice formation.

Apologies if we missed the point of your original comment. However, given that the primary signal of interest is the TNB record (for the reasons given in the revised manuscript), an alternative interpretation of the data that is true *except at TNB* is not very relevant. With regard to the other sites, none of the other locations have an observation in 2018, when the TNB record reaches a maximum, so we don't know the full extent of the "rebound" at those sites. Even so, values in 2017 exceed values in 2008 (there are no measurements in 2007, so we assume you mean the maximum in 2008, or 2006 at JT) at each site. In any case, analysis of "trends" based on selecting particular end points (e.g. the local maximum in 2008/2006) is not robust.

As the review states, the statement that *"the rebound just brings salinity back to the 2007 level"* does not apply to TNB, the primary signal of interest here.

*My other point, expressed elsewhere, was that the *absolute values of rates* of freshening (2007-2014) and rebound (post-2014) are "comparable" in that they are both much larger than the trend of the full record (1995-2018). So again, singling out the rebound "feature" in the overall trend, while ignoring the "rapid freshening" during 2007-2014, seems biased. Yes, the rebound *rate* is twice the 2008-2012 freshening rate, but the absolute values of both are an order of magnitude larger than the 23-year trend.*

In any signal with strong interannual variability, there will be short-term trends between maxima and minima that are larger than the long-term trend. We don't believe the relative magnitude of short-term trends is very informative (and hence we did not discuss them in the text). We only calculated them to address the comment in the second review.

As shown by the statistics presented in the response to the second reviews, at TNB the "rapid freshening" is consistent with the overall freshening trend (i.e. the trends calculated for different periods including and excluding the 2008-2014 interval are similar). In contrast, the increase in salinity from 2014 is anomalous relative to the record from 1995 to 2014. There is no "bias" in the presentation of our results. We do agree, as noted in the first point above, that there is strong interannual variability in the record and those variations provide clues to the physical mechanisms influencing the salinity of shelf waters.